# Comprehensive Analysis of the Prognostic Significance of the TRIM Family in the Context of TP53 Mutations in Cancers

**DOI:** 10.3390/cancers15153792

**Published:** 2023-07-26

**Authors:** Trung Vu, Annaliese Fowler, Nami McCarty

**Affiliations:** 1Center for Stem Cell and Regenerative Disease, Brown Foundation Institute of Molecular Medicine for the Prevention of Human Diseases (IMM), The University of Texas-Health Science Center at Houston, Houston, TX 77030, USA; trung.vu.1@uth.tmc.edu; 2The Department of Biomedical Engineering at Texas A&M University, Houston, TX 77030, USA; afowl@tamu.edu

**Keywords:** TP53, liver cancer, TRIM family, cell cycle

## Abstract

**Simple Summary:**

The p53 protein is an important tumor suppressor, and TP53 mutations are frequently low survival rates of cancer patients. Mutations in TP53 lead in loss of tumor suppressing functions which result in the development of tumors. Meanwhile, several tripartite motif (TRIM) proteins are known to regulate cell growth and cell cycle transition. However, the relationship between TRIM family genes and TP53 mutations in cancer remains unknown. In this study, we analyzed the links between TP53 mutations and TRIM family proteins and evaluated the role of TRIM family proteins in cancer patients with TP53 mutation. Our findings identified the TRIM family members that are highly expressed in TP53 mutant tumors and important for the cell growth in the context of TP53 mutations.

**Abstract:**

The p53 protein is an important tumor suppressor, and TP53 mutations are frequently associated with poor prognosis in various cancers. Mutations in TP53 result in a loss of p53 function and enhanced expression of cell cycle genes, contributing to the development and progression of cancer. Meanwhile, several tripartite motif (TRIM) proteins are known to regulate cell growth and cell cycle transition. However, the prognostic values between TP53 and TRIM family genes in cancer are unknown. In this study, we analyzed the relationship between the *TP53* mutations and TRIM family proteins and evaluated the prognostic significance of TRIM family proteins in cancer patients with P53 mutations. Our findings identified specific TRIM family members that are upregulated in *TP53* mutant tumors and are associated with the activation of genes related to a cell-cycle progression in the context of TP53 mutations.

## 1. Introduction

The TP53 gene is a high-frequency target of mutations in human cancers [1]. The p53 protein plays a critical role in regulating various cellular processes, including transcription, DNA synthesis and repair, cell cycle regulation, senescence, and apoptosis. Mutations in TP53 can disrupt these functions, leading to genetic instability and an increased risk of cancer development [2]. A comprehensive analysis of 12 major cancer types revealed that TP53 gene mutations are present in 42% of all cancers, with a minimum mutational rate of 20% in 10 out of the 12 cancer types studied [3]. Importantly, TP53 gene mutations are associated with poorer prognosis and unfavorable clinicopathological features [4]. 

The tripartite motif (TRIM) proteins belong to a large protein family, which are characterized by the presence of a conserved N-terminal RBCC module consisting of a Really Interesting New Gene (RING) domain, one or two B-Boxes (B1/B2) and a coiled-coil (CC) domain. So far, more than 80 TRIM family members have been identified in humans [5]. These TRIM proteins have been reported to be involved in a variety of important biological processes, including transcription, signal transduction, autophagy, cell proliferation, apoptosis, DNA repair, cell differentiation, stemness, antiviral infection, and immune response. The dysregulation of TRIM proteins has been linked to different diseases, including cancers, infectious diseases, developmental diseases, and neuropsychiatric disorders [5].

Recent studies have revealed the interplay between TRIM proteins and p53 levels and activity [6,7]. The crosstalk between the p53 signaling pathway and TRIM family proteins may play an important role in controlling cellular biological processes and impacting different types of cancer. In this study, we investigated TRIM family members that are upregulated in TP53 mutant tumors and whether they have prognostic significance in TP53 mutant tumors. Mutant TP53 cancers show enhanced expression of cell-cycle progression genes [8]. Our results demonstrated that various TRIM family members are upregulated in TP53 mutant cancers and are involved in the enhanced expression of genes related to cell-cycle progression in the context of TP53 mutations. The prognosis of liver cancers with TP53 mutations is strongly correlated with TRIM genes. Collectively, our data support that TRIM family members have a prognostic value in cancers, especially TP53 mutated tumors.

## 2. Materials and Methods

### 2.1. Cell Culture and Reagents

The human hepatocellular carcinoma cell line HepG2 was purchased from the ATCC. Cells were cultured in DMEM with 10% FBS. 30% Hydrogen peroxide was purchased from Sigma (St. Louis, MO, USA). Lentiviral vectors for shRNA against human TP53 were a gift from Dr. Dung-Fang Lee. 

### 2.2. RNA Extraction and RT-PCR

Total RNA was extracted using Direct-zol™ RNA MiniPrep (Zymo Research), and cDNA was synthesized using random hexamers and ImProm-II™ Reverse Transcription System (Promega. Madison, WI, USA). The synthesized cDNA was diluted 1:10 in nuclease-free water for PCR amplification. Quantitative real-time PCR was performed using SYBR Green MasterMix Plus (Thermo Scientific, Waltham, MA, USA) on ABI-7900 (Applied Biosystems, Waltham, MA, USA) with an initial temperature of 95 °C for 10 min, followed by 40 cycles of 15 s at 95 °C, 30 s at 60 °C, and 30 s at 72 °C. The primer sequences are described in Appendix A.

### 2.3. Immunoblotting

For immunoblotting, cells were collected and lysed in Laemmli sample buffer. Proteins were separated by SDS-PAGE and transferred onto the PVDF membrane. Anti-p53 (Santa Cruz Biotechnology, Dallas, TX, USA) and anti-β-actin (Sigma, Louis, MO, USA) antibodies were used to probe the membrane, followed by blotting with ECL reagent. β-actin’s expression levels were set as internal controls to normalize target protein expression.

### 2.4. Lentivirus Production and Transduction

HEK293FT cells were transfected with either shRNA against human TP53 or a lentivirus vector plasmid coupled with pMD2.G and psPAX2 by using calcium phosphate precipitation. Lentiviruses were collected 48 and 72 h post-transfection and concentrated via ultracentrifugation. For transduction, HepG2 cells were treated with 5 μg/mL polybrene and transduced with lentivirus. Medium containing lentivirus particles was replaced with regular medium after 24 h. At 96 h post-transduction, cells were selected with 1 μg/mL puromycin for 7 days, to generate stable cell lines.

### 2.5. Data Acquisition and Gene Set Enrichment Analysis

The study extracted normalized RNA-seq gene expression and overall survival data from The Cancer Genome Atlas (TCGA) for 12 cancer types via the University of California Santa Cruz (UCSC) Xena web portal. For each TRIM protein member, we grouped samples into ‘TRIM-high’ and ‘TRIM-low’ expressions using the median as the cutoff and performed differential gene expression (DGE) analysis. Gene set enrichment analysis (GSEA) was performed using GSEA v2.0 software (MSigDB https://www.gsea-msigdb.org/gsea/msigdb/ assessed on 24 November 2022) [8]. Enrichment results with a *p* < 0.05 and a false discovery rate (FDR) < 0.25 were considered statistically significant. This study also conducted enrichment analysis of KEGG pathways annotations of differentially expressed genes using the EnrichR online tool (http://amp.pharm.mssm.edu/Enrichr/ assessed on 27 November 2022) [9].

### 2.6. UALCAN

UALCAN (http://ualcan.path.uab.edu/analysis.html assessed on 17 October 2022) is a comprehensive, user-friendly, and interactive web resource for analyzing cancer OMICS data [10]. It is designed to provide easy access to publicly available cancer OMICS data, such as TCGA, MET500, and the clinical proteomic tumor analysis consortium (CPTAC). In this research, we used the function “Expression Analysis” of UALCAN to get the expression data of TRIM family members in 12 TCGA datasets. The Student’s *t*-test was used for analysis, and results with *p* < 0.05 were considered statistically significant.

### 2.7. Network Analyst

NetworkAnalyst (https://www.networkanalyst.ca/ assessed on 23 December 2022) [11] was applied to predict the miRNAs of the TRIM family members, and the results were presented by using Cytoscape 3.7.0 [12]. The predictions of miRNAs were based on the TarBase database [13].

### 2.8. Statistical Analysis

GraphPad Prism software 9.3.1 (GraphPad Software, LaJolla, CA, USA) was used for statistical analyses. Significant differences in mean values were evaluated by the Student’s *t*-test (unpaired, two-tailed), with *p* < 0.05 considered significant. Survival curves were compared by the log-rank (Mantel–Cox) test and presented with Kaplan–Meier curve.

## 3. Results

### 3.1. TP53 Mutation Is Linked to a Poorer Prognosis in Multiple Cancers

Multiple studies have investigated whether TP53 mutation affects survival, and some studies have found a correlation between TP53 mutation and a poorer prognosis [3]. In our study, we analyzed the impact of TP53 mutation on overall survival (Figure 1) using data from The Cancer Genome Atlas (TCGA) for 12 major TCGA cancers: bladder urothelial carcinoma (BLCA), breast invasive carcinoma (BRCA), colon adenocarcinoma (COAD), glioblastoma multiforme (GBM), head and neck squamous cell carcinoma (HNSC), low-grade glioma (LGG), liver hepatocellular carcinoma (LIHC), lung adenocarcinoma (LUAD), pancreatic adenocarcinoma (PAAD), sarcoma (SARC), stomach cancer (STAD), and uterine corpus endometrial carcinoma (UCEC). Our analysis found that TP53 mutation was significantly associated with overall survival in patients with COAD, HNSC, LIHC, LUAD, PAAD, and UCEC. These results suggest that TP53 mutation status affects the overall survival of patients with these types of cancer.

### 3.2. The Prognostic Value of TRIM Family Genes in the Context of TP53 Mutations

We examined the potential role of different TRIM expression levels in tumor progression in the context of *TP53* mutations. We analyzed the correlation between the expression levels of TRIM family members and overall survival in patients with 12 types of cancer with TP53 mutations. Patient samples with *TP53* mutations were divided into TRIM-Low or TRIM-high for each TRIM member using the median as the cutoff. We then used the Mantel-Cox test to analyze the overall survival of patients with TRIM-high compared with TRIM-low expression levels. Among the cancer types analyzed, LIHC had the highest number of TRIM family members, the overexpression of which is associated with poor prognosis in patients with TP53 mutations (Figure 2). Only one TRIM gene was relevant to the prognosis in BLCA (TRIM9), BRCA (TRIM46), PAAD (TRIM9), and UCEC (TRIM44). We did not observe any of TRIM family members with a prognostic value for patients with mutant TP53 in GBM, HNSC, LGG, STAD, and UCEC. 

We observed that elevated mRNA expression levels of TRIM3 [HR = 1.96 (1.01, 3.874), *p* = 3.2 × 10^−3^], TRIM6 [HR = 2.57 (1.125, 5.868), *p* = 0.02], TRIM11 [HR = 2.08 (0.98, 4.34), *p* = 0.02], TRIM24 [HR = 1.98 (0.9, 4.3), *p* = 0.035], TRIM25 [HR = 1.75 (1.046, 3.036), *p* = 0.037], TRIM28 [HR = 1.9 (1.04, 3.5), *p* = 0.031], TRIM32 [HR = 2.65 (1.065, 6.512), *p* = 0.02], TRIM44 [HR = 2.29 (1.042, 5.095), *p* = 0.008], TRIM45 [HR = 2.62 (1.065, 6.512), *p* = 0.02], and TRIM59 [HR = 1.95 (1.051, 3.55), *p* = 0.02] were associated with a shorter overall survival time in LIHC patients with TP53 mutations (Figure 3). However, we found no difference in the overall survival of TRIM-high patients with wild-type TP53 (Appendix A). 

In COAD patients with TP53 mutation, the mRNA levels of TRIM8 [HR = 1.7 (0.97, 3.01), *p* = 0.04], TRIM10 [HR = 1.77 (1.08, 3.12), *p* = 0.04], TRIM27 [HR = 2.47 (1.1, 5.37), *p* = 0.04], TRIM32 [HR = 1.8 (1.05,3.21), *p* = 0.03], TRIM39 [HR = 1.69 (0.93, 2.975), *p* = 0.045], and TRIM52 [HR = 1.637 (0.93, 2.8), *p* = 0.04] were associated with a poorer overall survival time (Appendix A), while in LUAD patients with TP53 mutations, TRIM6 [HR = 1.74 (1.1, 2.66), *p* = 0.007], TRIM7 [HR = 2.52 (1.619), *p* < 0.0001], TRIM15 [HR = 1.58 (1.01, 2.448), *p* = 0.03], TRIM28 [HR = 1.77 (1.1, 2.7), *p* = 0.006], and TRIM47 [HR = 1.83 (1.05, 3.21), *p* = 0.03] had prognostic values (Appendix A).

### 3.3. Identification of Upregulated TRIM Family Members in TP53 Mutant Liver Cancer Patients Compared to TP53 Wild-Type Patients

To investigate the clinical relevance of TRIMs in TP53 mutant tumors, we analyzed mRNA expression profiles of tumors from 12 types of cancer. For each type of cancer, we compared the transcriptional levels of TRIM family members in samples with TP53 mutations to those in wild-type TP53 samples (Figure 4). Importantly, our analysis found that the transcriptional levels of TRIM3, TRIM6, TRIM11, TRIM24, TRIM25, TRIM28, TRIM32, TRIM44, TRIM45, and TRIM59 were significantly elevated in TP53 LIHC samples compared to wild-type TP53 samples (Figure 5). Interestingly, the regulation of TRIM genes in TP53 mutant tumors is highly dependent on the type of cancer. Although TRIM3 is upregulated in TP53 mutant LIHC tumors, TRIM3 expression is downregulated in TP53 mutant BRCA, LGG, PAAD, and UCEC. TRIM6 is upregulated in TP53 mutant BLCA, HNSC, LIHC, and LUAD, while it is downregulated in BRCA and LGG. TRIM11 is upregulated in TP53 mutant GBM, LIHC, PAAD, and STAD, while it is downregulated in TP53 mutant BLCA, BRCA, and UCEC. TRIM24 is upregulated in TP53 mutant BLCA, BRCA, COAD, LIHC, and LUAD, while it is downregulated in TP53 mutant HNSC. TRIM25 is upregulated in TP53 mutant BLCA, BRCA, LIHC, and LUAD, while it is downregulated in TP53 mutant LGG and PAAD. TRIM28 is upregulated in TP53 mutant BLCA, BRCA, LGG, LIHC, LUAD, and UCEC, while it is downregulated in TP53 mutant HNSC. TRIM32 is upregulated in TP53 mutant BLCA, GBM, LIHC, while it is downregulated in TP53 mutant BRCA and UCEC. TRIM44 is upregulated in TP53 mutant BLCA, COAD, LIHC, and UCEC, while it is downregulated in TP53 mutant LGG. TRIM45 is upregulated in TP53 mutant BLCA, GBM, LGG, LIHC, LUAD, PAAD, SARC, and STAD, while it is downregulated in TP53 mutant BRCA. TRIM59 is upregulated in TP53 mutant BRCA, GBM, LIHC, LUAD, PAAD, and UCEC, while it is downregulated in TP53 mutant LGG (Figure 4).

On the other hand, except for TRIM39 and TRIM52, there is no difference in the expression levels of TRIM genes between mutant TP53 and wild-type TP53 COAD tumors (Appendix A). Except for TRIM28, there is no difference in the expression levels of TRIM genes between mutant TP53 and wild-type TP53 LUAD tumors (Appendix A).

Altogether, the results showed that the upregulation of the TRIM genes, which are associated with poor prognosis of patients with mutant TP53 tumors, was mainly observed in LIHC.

### 3.4. The Upregulation of TRIM Genes Due to TP53 Loss-of-Function Mutations

In liver cancer, mutations of p53 are 57% missense, 17% frameshift, 12% nonsense, 10% splice, and 4% other (Appendix A). While all cancer-relevant TP53 mutations probably abolish the tumor suppressive effects of wild-type p53, missense, but not null mutations, may potentially confer gain-of-function activities [14,15]. Therefore, we divide the TCGA LIHC samples into wild-type p53 tumors, p53-null tumors (harboring frameshift and nonsense mutations), and tumors with TP53 missense mutations to confirm whether the upregulation of TRIM genes is due to TP53 loss-of-function mutations. Patients with p53-null tumors showed increased expression levels of TRIM genes compared to patients with wild-type p53 (Appendix A). To confirm the effects of the loss-function of wild-type p53 on mRNA levels of TRIM genes, we knocked out TP53 expression with shRNA (sh-TP53) in the HepG2 hepatocellular carcinoma cell line. The HepG2 hepatoma cells express wild-type p53 and represent an ideal cell type to test p53 function. The downregulation of TP53 was confirmed by Western blot analysis (Figure 6A). We performed qRT-PCR with primers targeting TRIM genes. The results showed that, except for TRIM25 and TRIM 45, knockdown of TP53 significantly increased the mRNA levels of the indicated TRIM genes in HepG2 cells (Figure 6B). Activation of p53 by H_2_O_2_ also decreased the mRNA levels of several TRIM genes including TRIM3, TRIM11, TRIM24, TRIM25, TRIM28, TRIM32, and TRIM44 (Figure 6C). Altogether, the results showed that p53 partially suppressed the expression of several TRIM genes and that loss of p53 tumor suppressor function results in upregulation of these TRIM genes.

### 3.5. Correlations between TRIM Family Members and the Hallmarks of Cancer in TP53 Mutant and Wild-type Tumor Samples

We grouped samples into TRIM-high and -low expressions for each selected TRIM family member using the median as the cutoff. We used the DESeq R package to analyze differential gene expression and determine the gene signature associated with the high expression of TRIM family members in liver cancer. We conducted GSEA analyses of genes upregulated in mutant TP53 tumors with increased expression of TRIM family members. Previous studies have shown that GSEA performed on the upregulated genes in mutant TP53 tumors revealed highly significant enrichment of pathways directly involved in promoting cell cycle progression, including E2F target genes, signatures of G2M checkpoint control, and mitotic spindle [16]. Our GSEA results showed that high expression of TRIM25, TRIM28, TRIM32, TRIM44, TRIM45, and TRIM59 mRNAs are strongly associated with these signatures of TP53 mutations (Figure 7A). These signatures were not found to be enriched in wild-type TP53 tumors with increased expression of TRIM family members (Appendix A).

Our KEGG signaling pathway analysis revealed that genes related to cell cycle and DNA replication were significantly enriched in differentially expressed genes (DEGs) upregulated in mutant TP53 liver tumors with high expression of TRIM1 (Figure 7D), TRIM25 (Figure 7F), TRIM28 (Figure 7G), TRIM32 (Figure 7H), TRIM44 (Figure 7I), TRIM45 (Figure 7J), and TRIM59 (Figure 7K). Interestingly, we did not observe any significant enrichment of cell cycle regulation-related signatures in tumors with high expression levels of TRIM3, TRIM6, and TRIM24 (Figure 7A). However, signaling pathway analysis showed that high expression levels of TRIM3 (Figure 7B), TRIM6 (Figure 7C), and TRIM24 (Figure 7E) were associated with several signaling pathways, such as NF-kB and TNF signaling pathways.

A recent study established the mutant TP53 gene expression signature based on the aggregated expression of four genes, CDC20, PLK1, CENPA, and KIF2C. These four genes were almost invariably significantly overexpressed in all cancer types with mutant TP53. Moreover, these genes have been established as (a) targets of wild-type p53 repression, (b) promoters of cell cycle progression, (c) components of the G2/M checkpoint, and (d) established E2F targets (with the exception of CENPA) [16]. The correlation between the selected TRIM genes and these mutant TP53 signature genes was analyzed. All selected TRIM genes were positively correlated with CDC20, PLK1, CENPA, and KIF2C (Figure 8).

Taken together, these results suggest that these TRIM genes are correlated with TP53 signatures, and that TP53 mutation-mediated upregulation of TRIM family members may contribute to the enhanced expression of genes related to a cell-cycle progression in the context of TP53 mutations.

### 3.6. p53-Mediated Suppression of TRIM Family Members through the Induction of Specific miRNAs

Using NetworkAnalyst, we predicted potential miRNAs that may regulate TRIM family members (Figure 9A). Several of these miRNAs were found to be direct targets of p53 and were upregulated upon p53 activation [17]. These miRNAs contain p53 binding sites and are upregulated in cells with activated p53 (Figure 9B). Interestingly, several miRNAs were found to regulate multiple TRIM family members. For example, mir-34a-5p was found to regulate TRIM3, TRIM6, TRIM11, TRIM25, TRIM28, and TRIM32. Previous studies have shown that p53 binds to p53 response elements in miR-34a promoters and activates transcription of the miR-34 family [18]. In addition to miR-3a, other miRNAs that were found to be upregulated by p53 also potentially regulate the expression of multiple TRIM family members, such as mir-774-5p, let-7a-5p, let-7b-5p, mir-30a-5p, and mir-148a-3p. Furthermore, we analyzed the expression levels of these microRNAs in TCGA LIHC patient samples with missense and null TP53 mutations compared to patients with wild-type TP53. The results showed that the expression levels of mir-34, mir-30a, mir-24, mir-744, mir-423, and mir-16 are significantly increased in patients with null TP53 mutations (Appendix A). Our findings suggest an important role for miRNAs in the p53-mediated regulation of TRIM family members (Figure 9C).

## 4. Discussion

As a transcription factor, p53 plays a critical role in tumor suppression by binding to p53 DNA-binding elements in its target genes to regulate their expression. Through transcriptional regulation, p53 influences key biological processes, such as apoptosis, cell cycle arrest, senescence, DNA repair, cell metabolism, and antioxidant defense [19]. The majority of TP53 mutations occur in the DNA-binding elements of its target genes and subsequently affect the transcriptional activity of p53 [2]. 

Activation of the p53 tumor suppression function can lead to cell cycle arrest and suppression of a number of E2F target genes. The key mechanism of p53-mediated arrest is the transcriptional downregulation of many cell cycle genes. Notably, mutant TP53 cancer cells exhibit an enhanced expression of cell-cycle progression genes and S-phase-promoting E2F target genes [19].

Some TRIM proteins, when knocked down or silenced, cause an increase in the percentage of cells in G0/G1 and a reduction in the fraction of cells in S or G2 m phases. In particular, TRIM8, TRIM14, TRIM27, TRIM28, TRIM29, TRIM52, TRIM59, TRIM66, and TRIM68 are known to cause cell cycle arrest when depleted or silenced [20]. However, our findings suggest that the expression levels of these TRIM genes are not associated with TP53 mutations mediated activation of cell-cycle. In contrast, we found that high expression levels of TRIM11, TRIM24, TRIM25, TRIM28, TRIM32, TRIM44, TRIM45, and TRIM59 are specifically associated with gene signatures related to the cell cycle in patients with TP53 mutations. This observation is consistent with previous studies reporting that TRIM11 [21], TRIM24 [22], TRIM25 [23], TRIM28 [24], TRIM32 [25], TRIM44 [26], and TRIM59 [27] promote cell survival and growth of cancer cells. 

The interactions between p53 and miRNAs were first demonstrated through the identification of several miRNAs as direct targets of p53. As a transcription factor, p53 mainly exerts its function through its transcriptional regulation of protein-coding target genes to initiate cellular responses. Recent studies have revealed a novel mechanism for p53 in tumor suppression: p53 induces the expression of specific miRNAs, which show tumor suppressive functions. miRNAs are responsible for the downregulation of many mRNAs and proteins following p53 activation [28]. A recent study has identified several microRNAs that regulate TRIM8, TRIM14, TRIM27, TRIM28, TRIM29, TRIM52, TRIM59, TRIM66, and TRIM68 are upregulated by p53, thus highlighting the role of miRNAs in p53-mediated regulation of TRIM family members. 

Among the identified miRNAs, mir-34a-5p regulates TRIM25, TRIM11, TRIM32, TRIM28, TRIM3, and TRIM6. Previous studies have shown that p53 directly regulates the expression of the miR-34 family members, which consist of miR-34a, miR-34b, and miR-34c [18,29]. The miR-34 family is transcribed from two different genes, and p53 binds to p53 response elements in the miR-34a and miR-34b/c promoters to activate transcription of the family [30]. Ectopic expression of miR-34a promotes p53-mediated apoptosis, cell cycle arrest, and senescence, while inhibition of endogenous miR-34a strongly suppresses p53-dependent apoptosis in cells. Studies have shown that miR-34 family members directly target several genes involved in cell cycle regulation, cell proliferation, and survival [18]. Suppression of TRIM family members by miR-34a may be essential for the tumor suppressor function of p53. In addition to the miR-34 family, p53 directly regulates the transcriptional expression of several other miRNAs by binding to p53 response elements in their promoters.

## 5. Conclusions

In summary, our study demonstrated that upregulation of TRIM11, TRIM25, TRIM28, TRIM32, TRIM44, TRIM45, and TRIM59 is associated with TP53 mutations and may contribute to the activation of the cell cycle. We propose that these TRIM family members may play a crucial role in the TP53-mediated upregulation of cell-cycle-specific progression genes. Furthermore, our findings suggest that p53 may regulate TRIM family members through specific miRNAs. Future investigations are necessary to identify the precise miRNAs that modulate TRIM family members. Our results provide a novel perspective on the TRIM family functions in cancers.

## Figures and Tables

**Figure 1 cancers-15-03792-f001:**
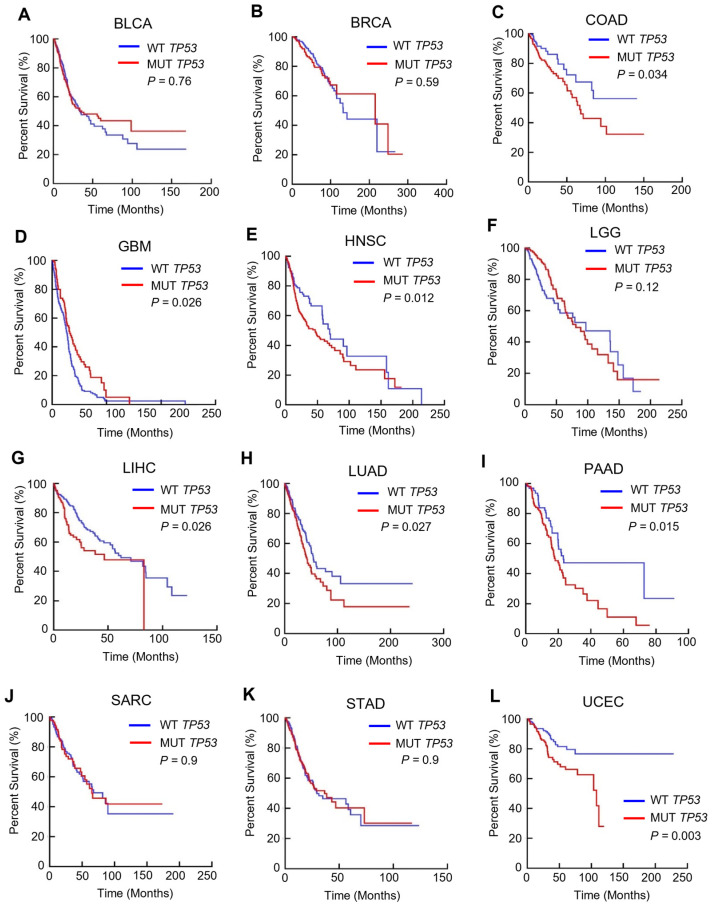
The effect of TP53 mutation status on overall survival of patients with 12 types of major cancers. Kaplan–Meier analyses were performed on the BCLA (**A**), BRCA (**B**), COAD (**C**), GBM (**D**), HNSC (**E**), LGG (**F**), LIHC (**G**), LUAD (**H**), PAAD (**I**), SARC (**J**), STAD (**K**), UCEC (**L**) overall survival data based on TP53 mutation status.

**Figure 2 cancers-15-03792-f002:**
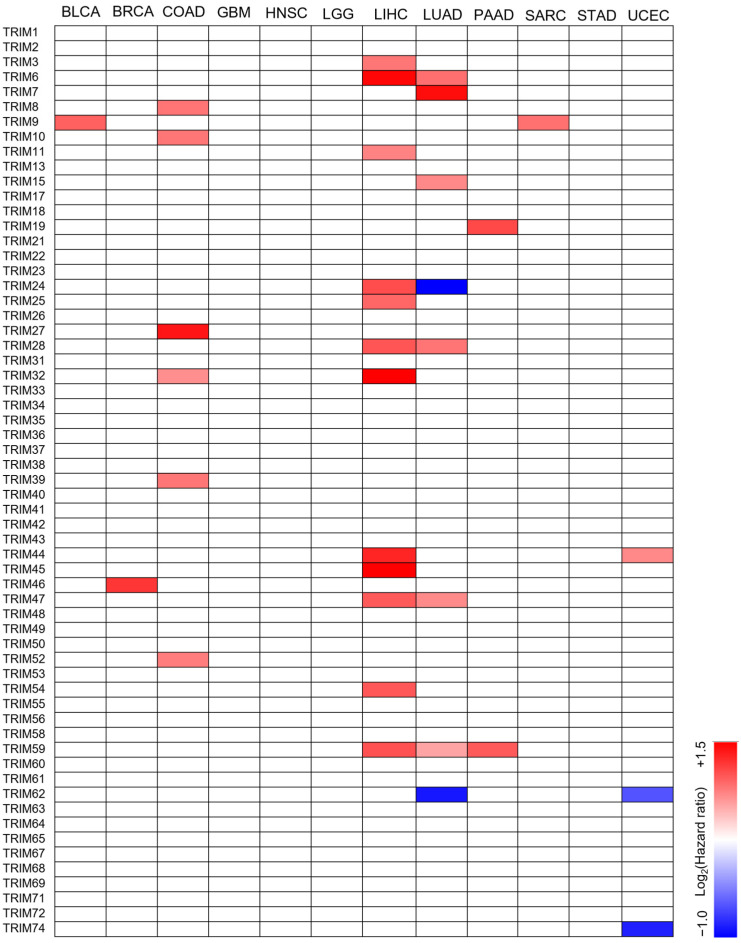
Hazard ratio (HR) heatmap representing associations between TRIM genes expression and overall survival (OS) of patients with TP53 mutant tumors. The heatmap is colored based on log2 HR. Only HRs with Wald test *p* value less than 0.05 are shown.

**Figure 3 cancers-15-03792-f003:**
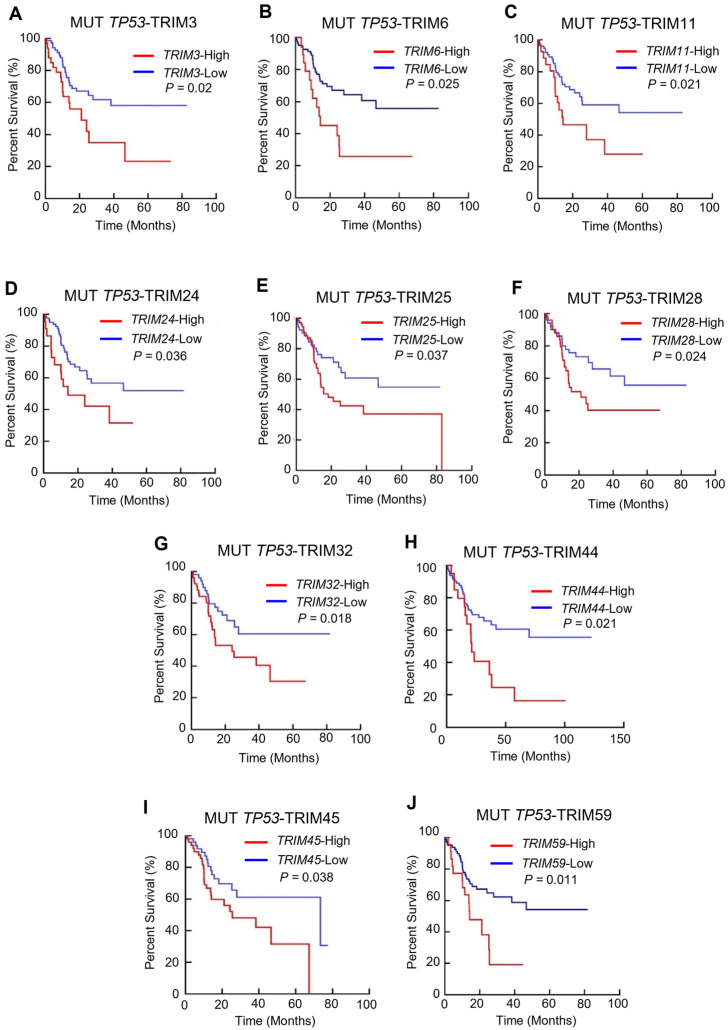
The prognostic values of the ten TRIM family members in the mutant (Mut) TP53 LIHC. (**A**–**J**) Kaplan–Meier analysis estimates of overall survival based on the expression levels of TRIM3 (**A**), TRIM6 (**B**), TRIM11 (**C**), TRIM24 (**D**), TRIM25 (**E**), TRIM28 (**F**), TRIM32 (**G**), TRIM44 (**H**), TRIM45 (**I**), and TRIM59 (**J**) in TCGA-LIHC patients.

**Figure 4 cancers-15-03792-f004:**
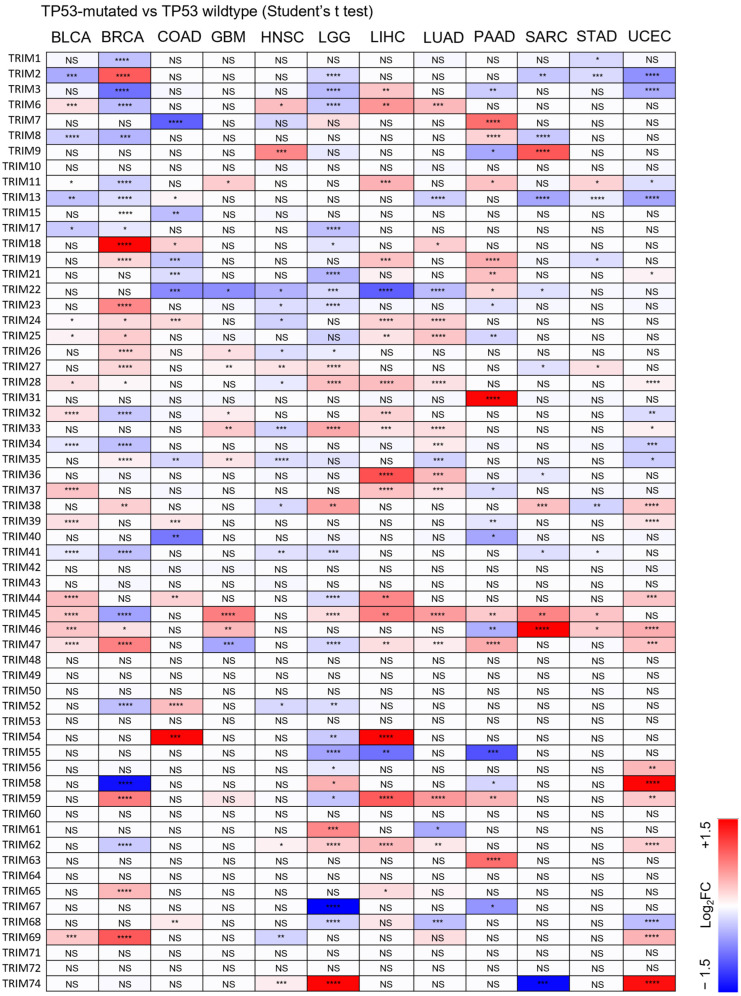
Heatmap showing the difference in mRNA levels of TRIM genes in TP53 mutant cancers compared to TP53 wild-type cancers. FC: fold change of mean gene expression levels (TP53 mutant cancers/TP53 wild-type cancers). * *p* < 0.05, ** *p* < 0.01, *** *p* <0.005, **** *p* < 0.001.

**Figure 5 cancers-15-03792-f005:**
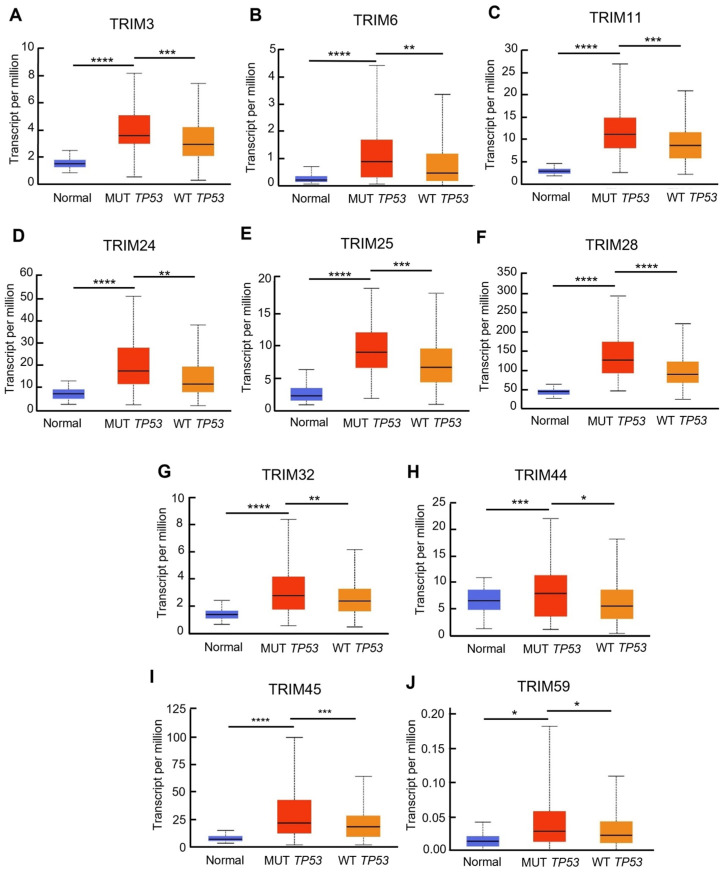
The association between mRNA levels of TRIM genes and TP53 mutation in LIHC. The transcriptional levels of TRIM3 (**A**), TRIM6 (**B**), TRIM11 (**C**), TRIM24 (**D**), TRIM25 (**E**), TRIM28 (**F**), TRIM32 (**G**), TRIM44 (**H**), TRIM45 (**I**), and TRIM59 (**J**) in normal liver tissues (Blue), mutant TP53 (Red), and wild-type TP53 (Orange) liver cancer tissues (UALCAN). * *p* < 0.05, ** *p* < 0.01, *** *p* <0.005, **** *p* < 0.001.

**Figure 6 cancers-15-03792-f006:**
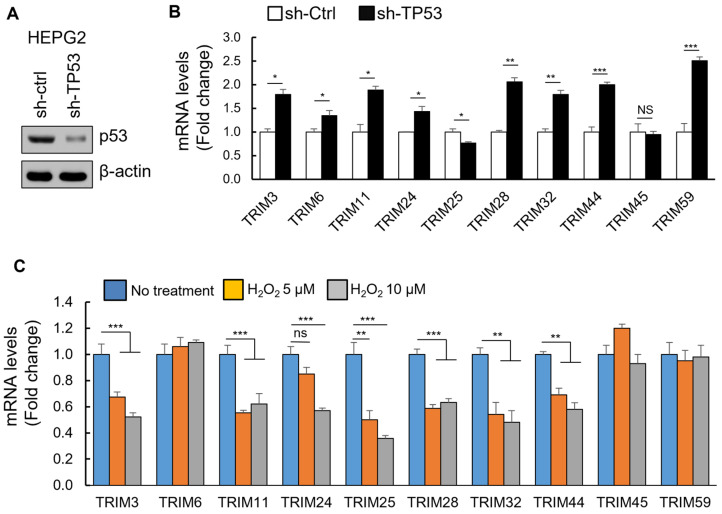
Knockdown of p53 increases endogenous TRIM genes in human hepatoma HepG2 cells. (**A**) Cells were infected with lentivirus containing TP53-targeting shRNA. Cell lysates were analyzed by Western blot for p53 by using anti-p53 antibody. B-actin was used as control. (**B**) Levels of mRNA in stable shTP53 HepG2 cells, as indicated, were determined by qRT-PCR and shown as relative fold changes using GAPDH as a loading control. (**C**) Cells were treated with H_2_O_2_ at indicated concentrations for 24 h. The mRNA levels of TRIM genes were determined by qRT-PCR. ns = not significant; * *p* < 0.05; ** *p* < 0.01, *** *p* < 0.005 when compared with the corresponding control. See Appendix A for original Western blot.

**Figure 7 cancers-15-03792-f007:**
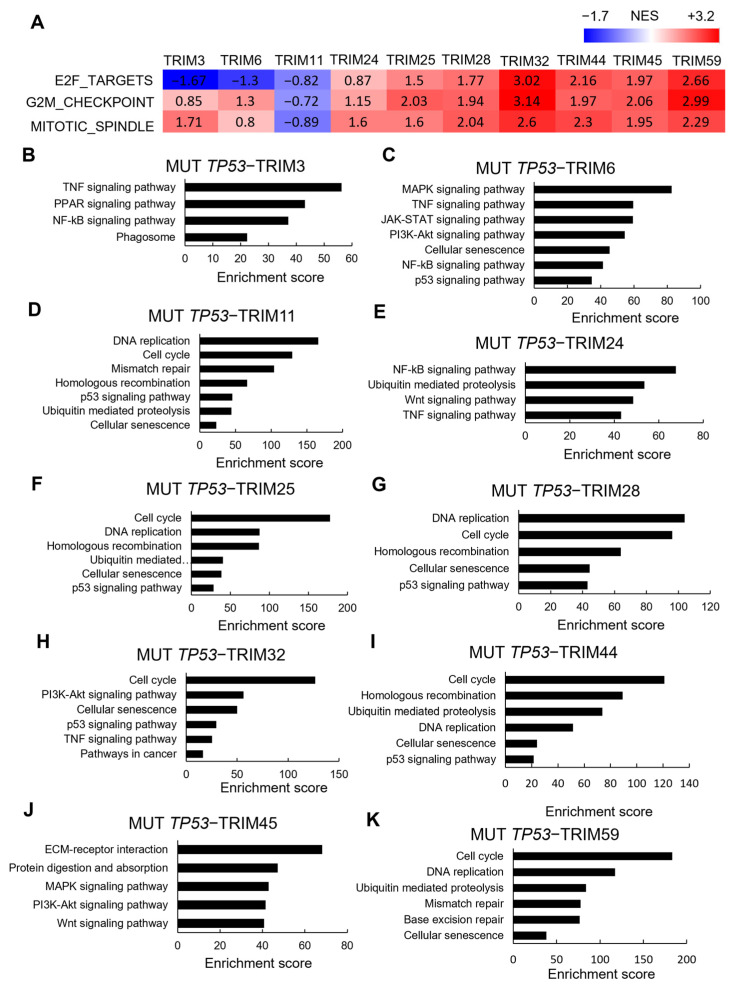
(**A**) Heatmaps of normalized enrichment scores (NES) calculated by GSEA against the Hallmark mitotic spindle, G2/M checkpoint control, and E2F target gene sets. (B-K) Genes upregulated in mutant TP53 LIHC samples with high expression levels of TRIM3 (**B**), TRIM6 (**C**), TRIM11 (**D**), TRIM24 (**E**), TRIM25 (**F**), TRIM28 (**G**), TRIM32 (**H**), TRIM44 (**I**), TRIM45 (**J**), and TRIM59 (**K**) were analyzed by the Enrichr web tool using the Kyoto Encyclopedia of Genes and Genomes (KEGG) libraries. The combined scores are depicted.

**Figure 8 cancers-15-03792-f008:**
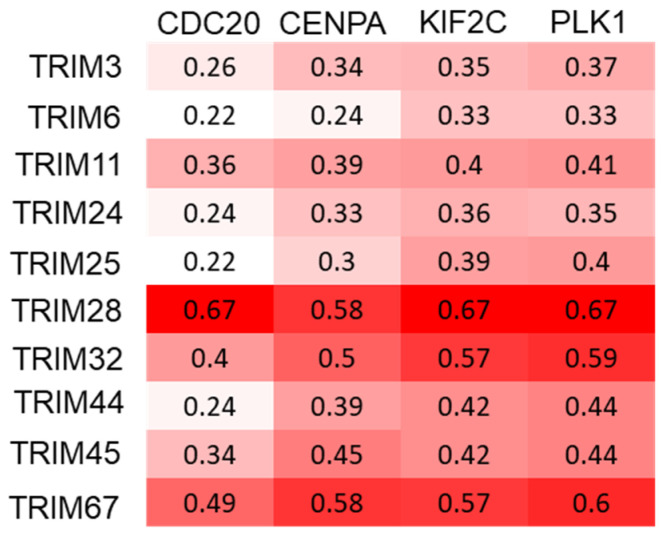
Overall correlations between the expressions of the ten TRIM family members and four mutant TP53 signature genes. Correlation coefficient R is displayed as color gradient, darker colors indicate higher correlations.

**Figure 9 cancers-15-03792-f009:**
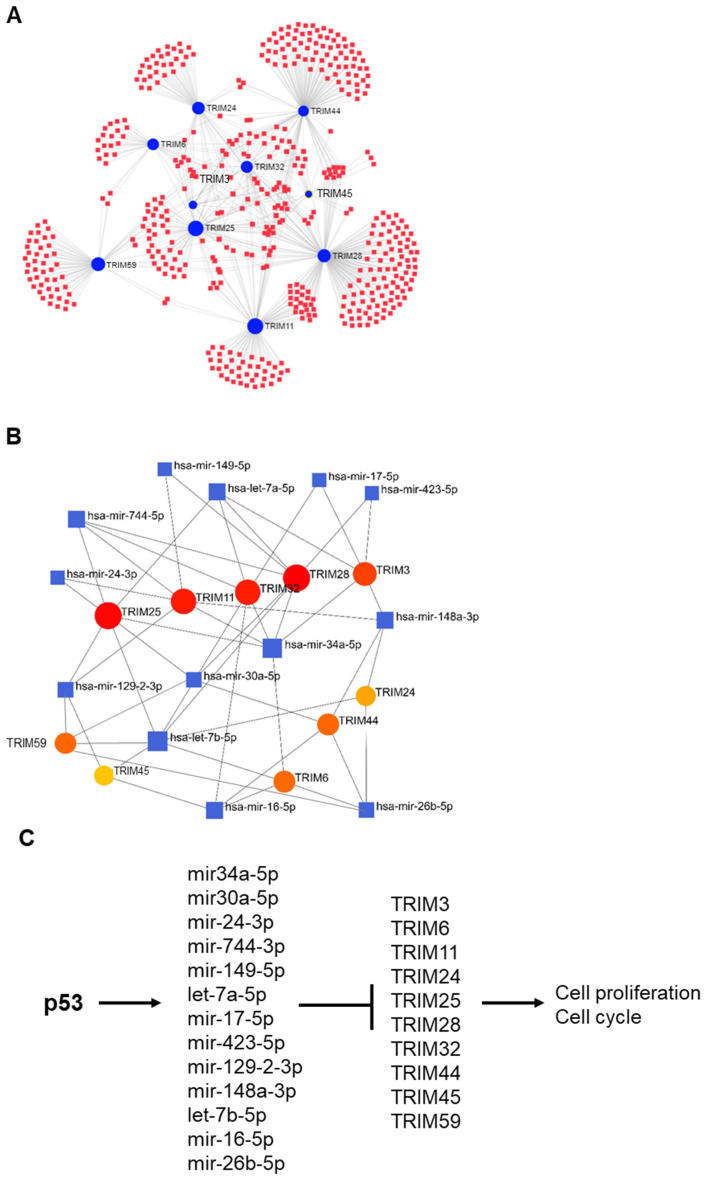
p53 suppresses TRIM family members through induction of specific miRNAs. (**A**) Curated TRIM-miRNA regulatory network constructed using Cytoscape, the red squares represent miRNAs, the blue circles represent TRIM genes. The size of the blue circles presents the number of connections they have to miRNAs. (**B**) The interaction network analysis of the p53-regulated miRNAs [17] and TRIM family members. The blue squares represent miRNAs. The size of blue squares represents the node degree, which is the number of connections they have to TRIM genes. The circles represent TRIM genes. The size and color present the number of connections they have to miRNAs (the number of connections decreases from red to yellow). (**C**) Schematic diagram of the proposed mechanism in which p53 suppresses TRIM genes through the upregulation of miRNAs.

## Data Availability

No new data were created in this study. Data sharing is not applicable to this article.

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
