# Peer review of "Comprehensive Analysis of the Prognostic Significance of the TRIM Family in the Context of TP53 Mutations in Cancers"

_cancers, 2023, doi:10.3390/cancers15153792_

Round 1

Reviewer 1 Report

This is a well-written manuscript titled “Comprehensive analysis of the prognostic significance of the 2 TRIM family in the context of TP53 mutations in cancers” that discusses the prognostic values between TP53 and TRIM family genes in cancer. In this present study, the author has analyzed the relationship between the TP53 mutations and TRIM family proteins and evaluated the prognostic potential of TRIM family proteins in cancer patients with P53 mutation. Collectively, the author identified specific TRIM family members whose expression is upregulated in TP53 mutant tumors and is associated with the activation of genes involved in a cell-cycle progression in context to TP53 mutations. This manuscript will be of good interest to readers working in the cancer field. The experiment design is very relevant to the addressed question in the manuscript. I recommend manuscript for the acceptance without any comments to be addressed. 

Author Response

Response: We would like to thank the reviewers for their positive and insightful comments.

Reviewer 2 Report

Authors analyze TCGA data to explore the correlation between p53 mutational status and expression of TRIM family proteins. In particular they search for TRIMs whose expression correlates with prognosis in mutp53 tumors. They focus on hepatocellular carcinoma (LIHC) and identify 10 TRIMs whose expression correlates with shorter patient survival in this tumor type.

Next, they confirm that the 10 TRIMs are more expressed in mut-p53 tumors with respect to wt-p53 tumors. This correlation is rather specific for LIHC, as the same genes are not upregulated in mut-p53 tumors of other types/origins. This implies that some cell- or tissue-specific parameter may be involved in the observed correlation.

GSEA analysis reveals that genes that are upregulated in TRIM-high tumors with mutp53 are enriched in cell-cycle and cancer-aggressiveness pathways.

Finally, they propose a possible mechanism for the observed correlation: the 10 TRIMs are potential targets of various micro-RNAs that have been reported to be transcriptionally regulated by wt-p53. Hence, they speculate that p53 mutation could reduce the levels of these miRNAs, thus favoring accumulation and overexpression of the TRIMs – with potentially pro-oncogenic effects.

This study is interesting, but has a number of flaws.

The first is that it is purely in silico. Some validation that TRIMs are indeed more expressed in mutp53 LIHC tumors (or cell models) is necessary. Ideally also at the protein level, as mRNA expression not always correlates with the encoded protein.

The second is that no evidence is provided that the TRIMs upregulated in mutp53 LIHC tumors may have a functional role in tumors aggressiveness. I understand that this may be a task left for future research, but without at least some indirect evidence in a cell model, the relevance of the work is greatly reduced. I expected at least an articulated speculation from literature data.

Finally, the hypothesis that the mechanism of TRIMs regulation may be mediated by p53-induced miRNAs is certainly plausible but merely speculative. It should be supported by a minimum of evidence. At least an in-silico analysis of p53-correlated expression of the miRNAs in LIHC would be required. 

I also have another general, quasi-philosophical point: from the methods it is not clear what criterion has been used to assign the p53 mutation status. Is it any type of mutation? Only homozygous or also heterozygous? This should have been stated more clearly.

In fact, we must take into consideration that the majority of p53 mutations are missense, and those mutations may confer to the mutp53 protein a number of oncogenic features (Gain Of Function). So LOF mutations may be different than GOF mutations – also in terms of expression of the TRIMs. Maybe this perspective could be taken in consideration and may reveal interesting results

A few specific points.

In the main text and in the figure legends, the authors often refer to 9 TRIMs, by I always see 10 TRIMs (?)

It is not clear how the enrichment was performed. The GSEA analysis is done on genes upregulated in TRIM-high p53-mutant LIHC samples. I presume they are compared with respect to TRIM-low mutp53 tumors.

But are the same Hallmarks also enriched in TRIM-high LIHC samples with wt-p53 ? This is a control that would be nice.

Also: this result implies that well-established mut-53 associated Hallmarks are not enriched in TRIM-low mut-p53 tumors? In other words, are the TRIMs upregulated only in those mut-p53 tumors that also upregulate cell-cycle genes?

I found no reference in the text to Fig S1 ?

The figure legends of Figure S5 and S6 retain references to liver tissue and liver cancer.

English is fine. Text has a few typos and minor errors - more frequent in the methods

Author Response

  1. The first is that it is purely in silico. Some validation that TRIMs are indeed more expressed in mutp53 LIHC tumors (or cell models) is necessary. Ideally also at the protein level, as mRNA expression does not always correlate with the encoded protein.

Response: We now include analyses of TRIM gene levels in P53 wild type liver cancer cells (Figure 6A) after silencing TP53. We also activated P53 by using H2O2 in wildtype TP53 liver cancer cells and analyzed the expression levels of different TRIM genes (Figure 6C). These data are in Figures 6A, 6B  and 6C. The results showed that downregulation of P53 in HepG2 liver cancer cells increased the expression levels of TRIM3, TRIM6, TRIM11, TRIM24, TRIM28, TRIM32, TRIM44, and TRIM59 (Figure 6B).

  1. The second is that no evidence is provided that the TRIMs upregulated in mutp53 LIHC tumors may have a functional role in tumors aggressiveness. I understand that this may be a task left for future research, but without at least some indirect evidence in a cell model, the relevance of the work is greatly reduced. I expected at least an articulated speculation from literature data.

Response: This is a great point. Previous studies showed that knockdown of TRIM11 (Liu J et al. 2018), TRIM24 (Zhu et al. 2018), TRIM25 (Liu et al. 2020), TRIM28 (Jin et al. 2017), TRIM32 (Cui et al. 2016), and TRIM59 (Ying  et al. 2020) in human hepatocellular carcinoma cell lines decreased its proliferation in vitro and tumor growth in vitro. We now include these articles in the discussion section.

  1. Finally, the hypothesis that the mechanism of TRIMs regulation may be mediated by p53-induced miRNAs is certainly plausible but merely speculative. It should be supported by a minimum of evidence. At least an in-silico analysis of p53-correlated expression of the miRNAs in LIHC would be required. 

Response: We have the analysis of p53-correlated expression of the miRNAs in LIHC patient samples in Figure S9. The results confirmed that mir-34, mir-30a, mir-24, mir-744, mir-433, and mir-16 are significantly increased in patients with null TP53 mutations (Figure S9).

  1. I also have another general, quasi-philosophical point: from the methods it is not clear what criterion has been used to assign the p53 mutation status. Is it any type of mutation? Only homozygous or also heterozygous? This should havef been stated more clearly.

In fact, we must take into consideration that the majority of p53 mutations are missense, and those mutations may confer to the mutp53 protein a number of oncogenic features (Gain Of Function). So LOF mutations may be different than GOF mutations – also in terms of expression of the TRIMs. Maybe this perspective could be taken in consideration and may reveal interesting results.

Response: We have now separated samples into two groups: those with TP53 missense mutations and those with TP53 null mutations. Subsequently, we reanalyzed the datasets. TP53 null mutations are specifically identified as loss of function mutants. These results revealed that

patients possessing either missense, and null TP53 mutations exhibit elevated mRNA levels of some TRIM genes (Figure S7).

  1. In the main text and in the figure legends, the authors often refer to 9 TRIMs, by I always see 10 TRIMs (?)

Response: We have corrected these figure legends throughout the manuscript.

  1. It is not clear how the enrichment was performed. The GSEA analysis is done on genes upregulated in TRIM-high p53-mutant LIHC samples. I presume they are compared with respect to TRIM-low mutp53 tumors. But are the same Hallmarks also enriched in TRIM-high LIHC samples with wt-p53? This is a control that would be nice.

Response: We have performed GSEA analysis in TRIM-high LIHC patients with wildtype TP53 as a control and include these data in Figure S8. We found that high expression levels of TRIM11, TRIM24, TRIM25, TRIM28, TRIM32, TRIM44, TRIM45, and TRIM59 are specifically associated with gene signatures related to the cell cycle in patients with TP53 mutations.

  1. Also: this result implies that well-established mut-53 associated Hallmarks are not enriched in TRIM-low mut-p53 tumors? In other words, are the TRIMs upregulated only in those mut-p53 tumors that also upregulate cell-cycle genes?

Response: Our results showed that TRIM genes are upregulated in mutant TP53 tumors and GSEA analysis showed that high expression of TRIM genes is associated with high expression of cell-cycle genes with mutant p53 background. This indicates that TRIM genes may function as a downstream signaling effector of mutant p53.   

  1. I found no reference in the text to Fig S1?

Response:  We have modified Figure S1 to Figure S6 and mentioned in the result session 3.4.

Round 2

Reviewer 2 Report

Authors have addressed most of the points raised, and provided a revised manuscript that includes an experiment showing that knockdown of wtp53 in HEPG2 upregulates the TRIMs and p53 activation by oxidative stress represses the TRIMs - at least at the RNA level (Fig 6); this supports the notion that p53 represses TRIMs expression.

The revised paper also reports the analysis of missense vs loss-of-function TP53 mutations (Fig S7), that reveals no relevant differences regarding TRIM gene expression, as well as regarding expression of TRIM-targeting miRNA (Fig S9). This result is in line with the authors’ model where p53 represses the TRIMs by transcriptionally upregulating a panel of miRNAs - this repression would be lost when p53 is either mutated or deleted (i.e. does no longer bind DNA)

These two additional data, plus some other adjustments, in my opinion have improved the paper. However, the revised manuscript still requires some adjustments and corrections. 

The methods, especially the new part, contain various errors that need to be fixed.

The concluding remark at line 215-216 is an overstatement, considering the indirect and limited evidence. Please re-write

Also the concluding sentence at lines 247-250 is too assertive, considering the evidence reported in Fig S8. The result is potentially interesting, and can be described better.

Fig 7A – one of the hallmarks is “mitotic-spindle”, NOT “cell-cycle”

Paragraph 3.6 – please fix reference to figure 9A

Figure 9 is described in a too superficial manner. Please explain in the legend the meaning of the graph in panel A (what are the tiny red squares?), the color-code in panel B, the evidence in support of p53-dependent regulation of miRNAs in panel C

Line 299 – the authors write miR-433 in the text and in the rebuttal letter; but is miR-423 in the figures. This should be fixed

Author Response

  1. The methods, especially the new part, contain various errors that need to be fixed.

Response: We have fixed the errors in the methods session.

  1. The concluding remark at line 215-216 is an overstatement, considering the indirect and limited evidence. Please re-write

Response: We have re-written the sentence.

  1. Also the concluding sentence at lines 247-250 is too assertive, considering the evidence reported in Fig S8. The result is potentially interesting, and can be described better.

Response: We have re-written the sentence to make it less assertive.

  1. Fig 7A – one of the hallmarks is “mitotic-spindle”, NOT “cell-cycle”

Response: We have fixed the error.

  1. Paragraph 3.6 – please fix reference to figure 9A

Response: We have fixed the error.  

  1. Figure 9 is described in a too superficial manner. Please explain in the legend the meaning of the graph in panel A (what are the tiny red squares?), the color-code in panel B, the evidence in support of p53-dependent regulation of miRNAs in panel C

Response: We have added more information in the legend of figure 9. For the evidence in support of p53-dependent regulation of miRNAs, we have included the reference from which we got the information about the miRNAs that were found to be upregulated by p53.

  1. Line 299 – the authors write miR-433 in the text and in the rebuttal letter; but is miR-423 in the figures. This should be fixed

Response: We have fixed the error.